# Comparative Analysis of Chemical Constituents of *Moringa oleifera* Leaves from China and India by Ultra-Performance Liquid Chromatography Coupled with Quadrupole-Time-Of-Flight Mass Spectrometry

**DOI:** 10.3390/molecules24050942

**Published:** 2019-03-07

**Authors:** Hongqiang Lin, Hailin Zhu, Jing Tan, Han Wang, Zhongyao Wang, Pingya Li, Chunfang Zhao, Jinping Liu

**Affiliations:** 1School of Pharmaceutical Sciences, Jilin University, Changchun 130021, China; linhq17@mails.jlu.edu.cn (H.L.); 13578965875@163.com (H.Z.); tanjing17@mails.jlu.edu.cn (J.T.); hanw17@mails.jlu.edu.cn (H.W.); zhongyao18@mails.jlu.edu.cn (Z.W.); lipy@jlu.edu.cn (P.L.); 2Research Center of Natural Drug, Jilin University, Changchun 130021, China

**Keywords:** *Moringa oleifera* leaves, analysis, UPLC-QTOF-MS, China, India

## Abstract

With the aim to discuss the similarities and differences of phytochemicals in *Moringa oleifera* leaves collected from China (CML) and India (IML) in mind, comparative ultra-performance liquid chromatography coupled with quadrupole-time-of-flight mass spectrometry (UPLC-QTOF-MS) analysis was performed in this study. A screening analysis based on a UNIFI platform was first carried out to discuss the similarities. Next, untargeted metabolomic analysis based on multivariate statistical analysis was performed to discover the differences. As a result, a total of 122 components, containing 118 shared constituents, were characterized from CML and IML. The structure types included flavonoids, alkaloids, glyosides, organic acids and organic acid esters, iridoids, lignans, and steroids, etc. For CML, 121 compounds were characterized; among these, 18 potential biomarkers with higher contents enabled differentiation from IML. For IML, 119 compounds were characterized; among these, 12 potential biomarkers with higher contents enabled differentiation from CML. It could be concluded that both CML and IML are rich in phytochemicals and that CML is similar to IML in the kinds of the compounds it contains, except for the significant differences in the contents of some compounds. This comprehensive phytochemical profile study provides a basis for explaining the effect of different growth environments on secondary metabolites and exists as a reference for further research into or applications of CML in China.

## 1. Introduction

*Moringa oleifera*, a herb native to India [1] which is also known as “miracle tree” or “the diamond in the plant”, has been widely cultivated throughout the world for its multiple uses such as its being a source of nutrients and a medical herb [2]. Most studies have focused on the leaves of the plant grown in India, Africa, or Madagascar [3,4]. The *Moringa oleifera* leaf (ML) have been proven to have antioxidant [5,6], anti-inflammatory [7,8], anticancer [9,10], anti-hypertensive [11], hypolipidemic [12], hypoglycemic [13,14], antimicrobial [15,16], and hepatoprotective [10,17] pharmacological activities. It has also been reported that ML contains many phytoconstituents such as flavonoids, alkaloids, steroids, saponins, glucosinolates, tannis, phenolic acids, and terpenes, etc. [18]. Certainly, its numerous pharmacological effects are due to the diversity of the phytochemicals in ML [19].

In China, as a complement to medicinal plant resources, *Moringa oleifera* was introduced from India in the 1960s and had been cultivated on a large scale in Guangdong Province, Yunnan Province, and other areas since then [20]. Additionally, ML was approved as a new food resource by the Chinese government in 2012 [21]. In China, relative research on extraction, preparation, and activity evaluation has been carried out recently, and there have been some achievements [22,23]. However, there has been a lack of profound research on the comprehensive screening and identification of the chemical constituents of ML grown in China. Furthermore, just as with other natural plants, *M. oleifera* ecotypes/cultivars differ from each other and can show many differences in leaf-mass production, growth performance, and secondary plant metabolite contents [24,25]. Therefore, with an aim to evaluate the similarities and the differences between the chemical constituents of Chinese *Moringa oleifera* leaf (CML) and Indian *Moringa oleifera* leaf (IML), a comparative analysis of the phytochemical composition of these two kinds of ML was performed in this study. On one hand, a comprehensive screening analysis of chemical components may be conducted to evaluate the similarity of CML to IML. During this section, a combination of ultra-high-performance liquid chromatography (UPLC) separation, quadrupole time-of-flight tandem mass spectrometry (QTOF-MS) detection and a UNIFI platform automated data process would be applied [26,27,28,29,30,31]. The accurate and specific mass could be provided by HR-MS when the coeluting constituents possess different *m*/*z* values. UNIFI might efficiently integrate data acquisition or mining and search libraries, and could generate reports using its comprehensive, simple, high throughput platform. The shared constituents of the Chinese and India *Moringa oleifera* leaves could be evaluated. On the other hand, with an aim to reveal the diversity of the metabolites, the untargeted metabolomics might be used to profile diverse classes of metabolites and compare the overall small-molecule metabolites of two kinds of samples [32]. This means a combination of UPLC separation, QTOF-MS detection, and multivariate statistical analyses, such as principal component analysis (PCA) and orthogonal projections to latent structures discriminant analysis (OPLS-DA), would be used to profile these two leaves.

The study in this paper comparatively analyzes the chemical constituents of *Moringa oleifera* leaves in China and India for the first time and determines the similarities and differences between these two items. Our data might support further research and the exploration of potential applications in China.

## 2. Materials and Methods

### 2.1. Materials and Reagents

CML and IML were collected from their respective cultivation areas or purchased from herbal markets in China or India (Table 1). The identity of the *Moringa oleifera* leaf was confirmed by the authors and the corresponding voucher specimens were deposited in the Research Center of Natural Drug, School of Pharmaceutical Sciences, Jilin University, China.

Methanol and acetonitrile (Fisher Chemical Company, USA) were used as they were suitable for UPLC-MS. Deionized water was purified using a Millipore water purification system (Millipore, Billerica, MA, USA). Formic acid for UPLC was purchased from the Sigma-Aldrich Company. All other chemicals were of analytical grade.

Standard compounds *α*-maltose, adenosine, catechin, chlorogenic acid, rutin, quercetin, kaempferol, caffeic acid, oleic acid, epicatechin, hyperoside, kaempferol-3-*O*-rutinoside, isorhamnetin, isorhamnetin-3-*O*-rutinoside, luteolin, scutellarein, methyl palmitate, ricinoleic acid, linolenic acid, dibutyl sebacate, eugenol, azelaic acid, (−)-epiafzelechin, methyl myristate, and 2′-hydroxygenistein were purchased from the National Institutes for Food and Drug Control (Beijing, China). Other reference compounds including parinaric acid, quinic acid, and 1,3-dicaffeoylquinic acid were purchased from Beijing Zhongke Quality Inspection Biotechnology Co., Ltd. (Beijing, China).

### 2.2. Sample Preparation and Extraction

Stalks were removed and the leaves air-dried, grinded, and sieved (Chinese National Standard Sieve No. 3, R40/3 series) to obtain a homogeneous powder. Then, the powder (1.0 g) was extracted with 80% methanol (1.0 L) at 80 °C thrice (for 3 h each time). After being filtered, the extraction solution was combined, concentrated, and evaporated to dryness. The desiccated extractions (all approximately 15 mg) were finally dissolved and diluted with 80% methanol 10.0 mL. The solution was filtered with a syringe filter (0.22 μm) and then injected into the UPLC system. Additionally, to ensure the suitability and stability consistency of MS analysis, a quality control (QC) sample was prepared by pooling the same volume (50 μL) from every sample. Through the whole worklist, 3 QC injections were performed randomly. The volume injected for the samples and QC was 2 μL for each run.

### 2.3. UPLC-QTOF-MS^E^

UPLC-QTOF-MS^E^ analysis was performed on a Waters Xevo G2-XS QTOF mass spectrometer (Waters Co., Milford, MA, USA) equipped with a UPLC system through an electrospray ionization (ESI) interface. Chromatographic separation was performed on an ACQUITY UPLC BEH C_18_ (100 mm × 2.1 mm, 1.7 μm) column provided by Waters Corporation. The mobile phases were composed of eluent A (0.1% formic acid in water, *v*/*v*) and eluent B (0.1% formic acid in acetonitrile, *v*/*v*) with flow rate of 0.4 mL/min. The elution conditions applied were: 0–2 min, 10% B; 2–26 min, 10–100% B; 26–29 min, 100% B; 29–29.1 min, 100–10% B; 29.1–32 min, 10% B. Mixtures of 90/10 and 10/90 water/acetonitrile were used as the weak wash solvent and the strong wash solvent, respectively. The temperatures of the column and autosampler were 30°C and 15 °C, respectively. The mass spectrum was acquired from 100 to 1500 Da in MS^E^ mode. The positive mode conditions were as follows: capillary voltage, 2.6 kV; source temperature, 150 °C; cone voltage, 40 V; cone gas flow, 50 L/h; desolvation temperature, 400 °C; desolvation gas flow, 800 L/h. Negative mode conditions were identical to the positive mode conditions except for the capillary voltage (2.2 kV). During a single LC run, data acquisition was performed via the mass spectrometer by rapidly switching from a low collision energy (CE) scan to a high-CE scan in MS^E^ mode. The collision energy of low energy function was set to 6 V while the ramp collision energy of high energy function was set to 20~40 V. Leucine enkephalin (LE) (*m*/*z* 554.2615 in ESI^−^ mode and 556.2771 in ESI^+^ mode), the external reference of Lock Spray™, was infused at a constant flow of 10 μL/min. During acquisition, data were collected in continuum mode for the screening analysis and in centroid mode for the metabolomics analysis. Masslynx™ V4.1 workstation (Waters, Manchester, UK) was used to record the data.

### 2.4. Screening Analysis of Components of CML and IML by UNIFI Platform

To quickly identify the chemical compounds, the MS raw data, compressed with Waters Compression and Archival Tool v1.10, was automatedly screened and identified using the streamlined workflow of UNIFI 1.7.0 software (Waters, Manchester, UK) [30,31,32,33]. The parameters were as follows: for 2D peak detection, 200 was set as the minimum peak area; for 3D peak detection, the peak intensities of low energy and high energy were set as over 1000 and over 200 counts, respectively; mass error in the range of ±5 ppm was set for identified compounds; retention time in the range of ±0.1 min was allowed to match the reference substance. Generated predicted fragments from the structure were identified as the matching compounds. Negative adducts containing +COOH and -H and positive adducts containing +H and +Na were selected in the analysis. Leucine enkaplin was selected as the reference compound, and [M − H]^−^ 554.2620 was used for the negative ion and [M + H]^+^ 556.2766 for the positive ion. Components were further verified by comparing reference substances with retention time and by comparing characteristic MS fragmentation patterns in the literature. The chemical information database used for the components was as follows: besides the in-house Traditional Medicine Library in the Waters UNIFI platform, the investigation of chemical constituents was conducted systematically. A self-built database of compounds that were reported in ML was established by searching online databases or internet search engines such as PubMed, Full-Text Database (CNKI), ChemSpider, Web of Science, and Medline. Chemical information including the component name, structures of the components, and molecular formula were available from the database.

### 2.5. Metabonomics Analysis of CML and IML

The raw data were processed for alignment, deconvolution, and data reduction, etc., with MarkerLynx XS V4.1 software (Waters, Milford, CT, USA) [34]. A Markerlynx processing method was first created, and its main parameters included: retention time (RT) range 0~26 min, minimum intensity 5%, mass range 100~1500 Da, mass tolerance 0.10, mass window 0.10, marker intensity threshold 2000 counts, retention time window 0.20, and noise elimination level 6. After processing the data, the results were able to be shown in Extended Statistics (XS) Viewer. *m*/*z*-RT pairs with corresponding intensities for all the detected peaks from each data file were listed. The same values of RT and *m*/*z* in different batches of samples were regarded as the same component. Furthermore, multivariate statistical analysis was performed. Firstly, PCA was used to show the pattern recognition and maximum variation aiming to obtain the overview and classification. Secondly, OPLS-DA in ESI^+^ and ESI^−^ modes was performed in order to get the maximum separation between the CML and IML groups and to explore the potential chemical markers that contribute to the differences. Then, S-plots were created to provide visualization of the OPLS-DA predictive component loading to facilitate model interpretation. Meanwhile, the use of variable importance for the projection (VIP) was helpful in screening the different components, and metabolites with VIP value > 1.0 and *p*-value below 0.05 were considered as potential markers [32]. In addition, permutation testing was performed to provide reference distributions of the R^2^/Q^2^ values that could indicate statistical significance [35,36]. Simca 15.0 software (Umetrics, Malmö, Sweden) was used to show the analysis results [33,35].

## 3. Results

### 3.1. Identification of Components from CML and IML Based on the UNIFI Platform

As a result of screening analysis, a total of 122 compounds were identified or tentatively characterized in both ESI^+^ and ESI^−^ mode from CML and IML. There were 118 shared constituents identified in CML and IML. More specifically, 121 and 119 compounds were characterized from CML and IML, respectively (Table 2). Both of the two types of *Moringa oleifera* leaves are rich in natural components with various structural patterns, including flavonoids, alkaloids, glyosides, organic acids and organic acid esters, iridoids, lignans, and steroids, etc. Base peak intensity (BPI) chromatograms marked with the number of compounds are shown in Figure 1. The chemical structures of the compounds are summarized in Figure 2. 

### 3.2. Diversity Evaluation of CML and IML Using Metabolomics Analysis

The QC injections were clustered tightly in PCA, indicating a satisfactory stability of the system. According to their common spectral characteristics, the PCA 2D plots of the samples from CML and IML groups were able to be easily classified within two clusters (Figure 3). The CML and IML samples were clearly separated, indicating that these two samples could be easily differentiated.

In order to evaluate the differences between the leaves in the two areas, OPLS-DA score plot, S-plot, permutation test, and variable importance in the projection values were obtained to understand which variables were responsible for this sample separation [72]. After OPLS-DA plots (Figure 4a and Figure 5a) in both ESI^+^ and ESI^−^ modes were performed, the maximum separation between the CML and IML groups was available. With sufficient permutation testing, the lines of grouping samples were significantly located underneath the random sampling lines (Figure 4b and Figure 5b), which indicates a definite validity for the following characteristic metabolites biomarkers identification. S-plots were then created to explore the potential chemical markers that contributed to the differences. Based on *p* values (*p* < 0.05) and VIP values (VIP > 1) [26,30] from univariate statistical analysis, 30 robust known chemical markers enabling differentiation between CML and IML were marked and listed (Figure 4c and Figure 5c and Table 2). Additionally, a heatmap was generated from these chemical markers in order to systematically evaluate the markers (Figure 6), which visually showed the intensities of potential chemical markers between the two samples.

## 4. Discussion

Via the screening analysis, 121 and 119 compounds were characterized in CML and IML, respectively. As the results show, 93 compounds were identified in negative mode and 29 compounds were identified in positive mode. From the BPI chromatograms, it seems that the negative ionization mode was better than the positive mode based on the quantity and the responses of the identified compounds. However, it was still necessary to have run the positive mode because some compounds showed better responses in this mode than in the negative mode. The results also showed that both these ML areas are rich in natural components. It has been reported that there is high flavonoid content (presenting in flavanol and glycoside forms) in *M. oleifera* leaves [4,18]. In this study, flavonoids were also the main chemical composition. Besides the most common flavonoids, 36 flavonoids, such as apigenin-8-C-glucoside, quercetin 3-*O*-*β*-d-glucopy-ranoside, kaempferol-7-*O*-α-l-rhamnoside, and 5, 7, 2′, 5′-tetrahydroxyflavone, were identified or tentatively characterized in *M. oleifera* leaves for the first time. Moreover, isothiocyanates have become a major topic of research interest regarding *Moringa* for their various biological activities [18]. In our study, there were 4 isothiocyanates which were found both in IML and CML. A total of 118 compounds were shared constituents in CML and IML, which means that they were similar in terms of the kinds of compound contained. This comprehensive phytochemical profile study has revealed the structural diversity of secondary metabolites and the similar patterns within CML and IML.

Furthermore, in nontargeted metabolomic analysis, when taking the contents of the constituents into account, it was found that there indeed existed differences between CML and IML. Thirty robust known biomarkers enabling this differentiation were discovered. These are able to illustrate the differences between CML and IML and provide a basis for explaining the effect of different growth environments on secondary metabolites. With CML, there are 18 potential biomarkers, including seven flavonoids (14, 33, 55, 63, 74, 79, and 84), five organic acids and organic acid esters (30, 38, 80, 115, and 116), two glyosides (12 and 29), and four others (69, 93, 121, and 122). Among these biomarkers, compounds 14, 33, and 38 were detected only in CML under experimental conditions, and the others’ contents in CML were greater than those in IML. Among these potential biomarkers, components 14, 33, 55, 74, 79, 30, and 80 were identified or tentatively characterized in *M. oleifera* leaves for the first time. It has been reported that *M. oleifera* leaves which originate from China have the maximum antioxidant activity when compared alongside those from Faisalabad, Multan, and India [73]. As is known, biological activity is caused by the high contents of phytochemicals. Correlation studies between potential markers and biological activities should be performed in the future. For IML, there are 12 potential biomarkers, including six flavonoids (50, 51, 59, 62, 78, and 82), three organic acids and organic acid esters (81, 97, and 119), one glyoside (100), one alkaloid (104), and one lignan (60). Among these, compound 82 was detected only in IML under experimental conditions, and the other 11 compounds’ contents were greater in IML than those in CML.

Based on the above results, it could be concluded that some of the secondary plant metabolite contents of CML and IML differ from each other. This is just as it is with other natural plants.

In summary, a total of 122 components, including 118 shared constituents, were characterized from CML and IML. For CML, 121 compounds were characterized, and among these, 18 potential biomarkers with higher contents enabled differentiation from IML. For IML, 119 compounds were characterized, and among these, 12 potential biomarkers with higher contents enabled differentiation from CML.

Even so, several unresolved issues still remain. For example, in the future, potential chemical markers’ and identified compounds’ pharmacological activities should be screened. In addition, there are still some unidentified components, despite 122 compounds being identified, as shown in the BPI chromatograms. Further research should be performed on these unknown components.

## 5. Conclusions

In this study, 121 and 119 chemical compounds, including 118 shared constituents, were respectively identified or tentatively characterized from CML and IML by combining UPLC-QTOF-MS and a UNIFI platform. Both CML and IML, which originate from two separate countries, are rich in phytochemicals and are similar in the kinds of compounds they contain. Moreover, a metabolomics study based on UPLC-QTOF-MS combined with multivariate statistical analysis has shown the significant differences in the contents of an amount of the compounds in these two accessions. A total of 30 robust known biomarkers enabling differentiation were discovered. For CML and IML, 18 and 12 potential biomarkers were identified, respectively. This study provides further data to make up for the deficient amount of study performed on the chemical constituents of *Moringa oleifera* leaves and can help with planning strategies focused on the proper utilization of this resource, as well as providing a reference for the further application of CML in China.

## Figures and Tables

**Figure 1 molecules-24-00942-f001:**
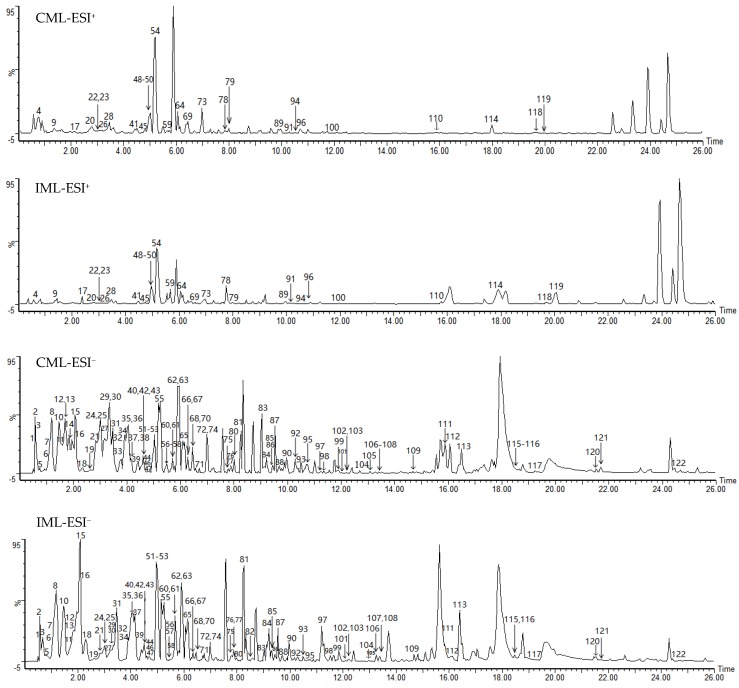
The representative base peak intensity (BPI) chromatograms of CML and IML in ESI^+^ and ESI^−^ modes, where ESI is electrospray ionization.

**Figure 2 molecules-24-00942-f002:**
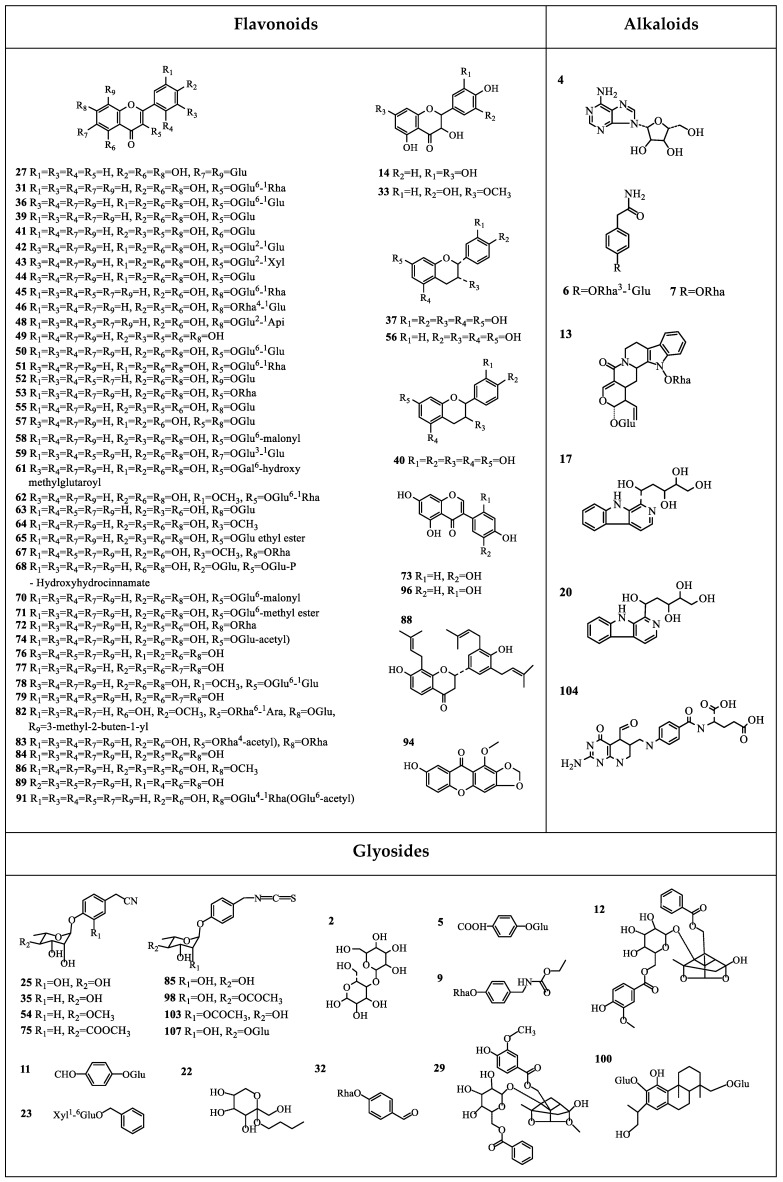
Chemical structures of compounds identified in CML and IML.

**Figure 3 molecules-24-00942-f003:**
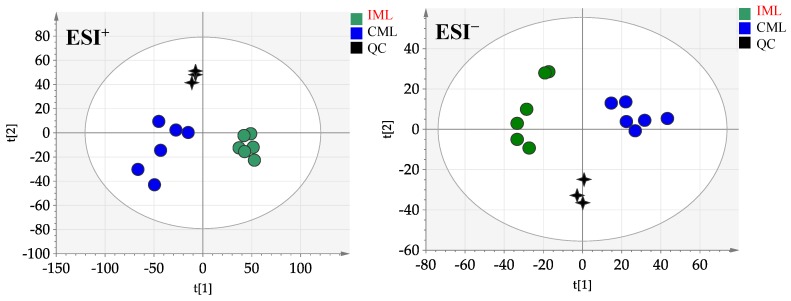
The principal component analysis (PCA) of CML and IML in ESI^+^ mode and ESI^−^ mode.

**Figure 4 molecules-24-00942-f004:**
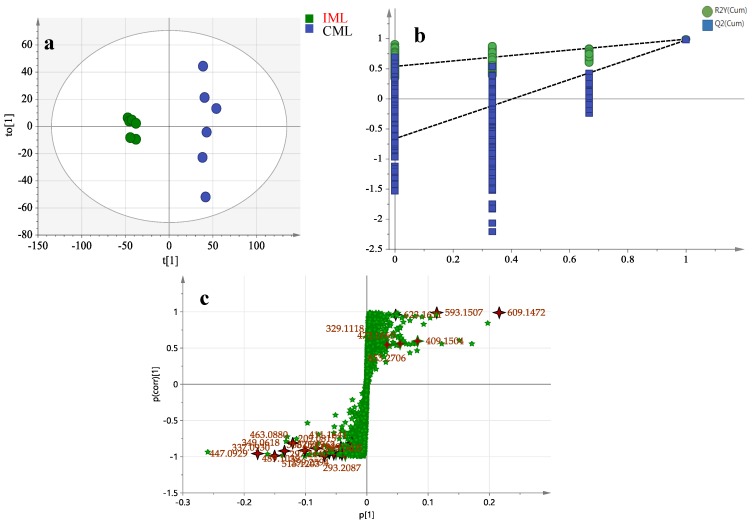
Orthogonal partial least squares discriminant analysis (OPLS-DA) (**a**), permutation tests, (**b**) and S-plot (**c**) in ESI^−^ mode.

**Figure 5 molecules-24-00942-f005:**
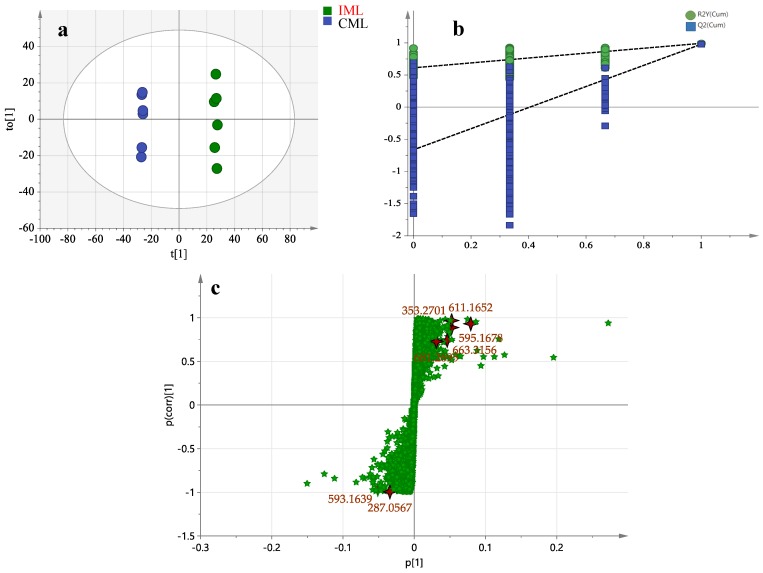
OPLS-DA (**a**), permutation tests, (**b**) and S-plot (**c**) in ESI^+^ mode.

**Figure 6 molecules-24-00942-f006:**
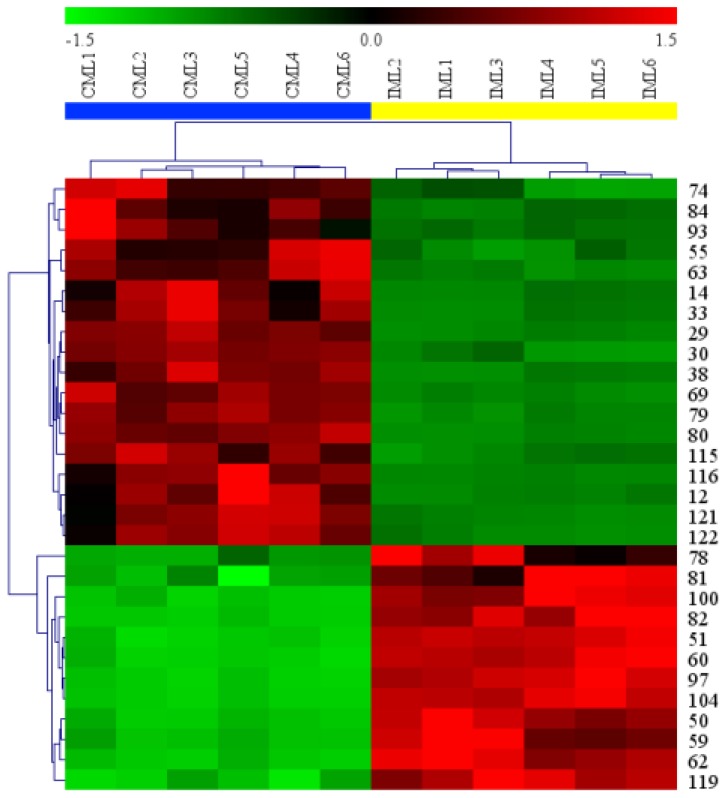
Heatmap visualizing the intensities of potential chemical markers.

**Table 1 molecules-24-00942-t001:** The list of the tested samples from China and India. Legend: CML, Chinese *Moringa oleifera* leaf; IML, Indian *Moringa oleifera* leaf.

Species	Sample No.	Source	Collection Time
CML	1	Pu‘er City, Yunnan Province, China; market	November 2017
	2	Xishuangbanna City, Yunnan Province, China; field	March 2018
	3	Shaoguan City, Guangdong Province, China; market	January 2017
	4	Guangzhou City, Guangdong Province, China; field	December 2017
	5	Danzhou City, Hainan Province, China; market	January 2018
	6	Changjiang City, Hainan Province, China; market	March 2017
IML	1	Howrah, India; market	December 2017
	2	Howrah, India; market	November 2017
	3	Tamil Nadu, India; market	February 2018
	4	Tamil Nadu, India; market	March 2018
	5	Maharastra, India; market	January 2018
	6	Maharastra, India; market	January 2017

**Table 2 molecules-24-00942-t002:** Compounds identified from CML and IML by ultra-high-performance liquid chromatography quadrupole time-of-flight tandem mass spectrometry UPLC-QTOF-MS^E.^

No.	Retention Time (RT) (min)	Formula	Calculated Mass (Da)	Theoretical Mass (Da)	Mass Error (ppm)	MS^E^ Fragmentation	Identification	Sources	Ref.
1	0.59	C_7_H_12_O_6_	192.0629	192.0634	−2.6	191.0542[M − H]^−^, 173.0432[M-H-H_2_O]^−^, 145.0516[M-H-HCOOH]^−^, 137.0232[M-H-3H_2_O]^−^, 127.0401[M-H-H_2_O-HCOOH]^−^	Quinic acid	CML, IML	s
2	0.60	C_12_H_22_O_11_	342.1161	342.1162	−0.3	387.1143[M + HCOO]^−^, 179.0554[M-H-Glu]^−^	*α*-Maltose	CML, IML	s
3	0.62	C_16_H_18_O_9_	354.0968	354.0951	4.8	353.0895[M − H]^−^, 335.0896[M-H-H_2_O]^−^, 190.0544[M-H-C_9_H_7_O_3_]^−^, 190.0391[M-H-3H_2_O-C_6_H_5_O_2_]^−^, 143.0346[M-H-HCOOH-C_9_H_8_O_3_]^−^	Cryptochlorogenic acid	CML, IML	[37]
4	0.72	C_10_H_13_N_5_O_4_	267.0968	267.0968	0.0	268.1041[M + H]^+^, 187.0620[M + H-C_3_H_3_N_3_]^+^, 161.0744[M + H-C_4_H_3_N_4_]^+^, 136.0612[M + H-Rib]^+^	Adenosine	CML, IML	s
5	0.80	C_13_H_16_O_8_	300.0836	300.0845	−2.9	299.0764[M − H]^−^, 178.0632[M-H-C_7_H_5_O_2_]^−^, 135.0231[M-H-Glu]^−^, 89.0347[M-H-Glu-HCOOH]^−^	Benzoic acid 4-*O*-*β*-glucoside	CML, IML	[18]
6	1.03	C_20_H_29_NO_11_	459.1739	459.1741	−0.2	504.1721[M + HCOO]^−^, 427.1487[M-H-NH_2_-CH_3_]^−^, 307.0995[M-H-C_8_H_9_NO_2_]^−^, 279.1081[M-H-Glu]^−^, 150.0546[M-H-2Glu]^−^	3 ′′-*O*-*β-*d-glucopyranosyl derivatives (marumoside B)	CML, IML	[38]
7	1.04	C_14_H_19_NO_6_	297.1210	297.1212	−0.7	342.1192[M + HCOO]^−^, 262.0758[M-H-H_2_O-NH_2_]^−^, 149.0546[M-H-Rha]^−^, 105.0430[M-H-Rha-CONH_2_]^−^	4 ′-Hydroxyphenylethanamide-*α-*l-rhamnopyranoside (marumoside A)	CML, IML	[38]
8	1.18	C_16_H_18_O_9_	354.0942	354.0951	−2.4	353.0869[M − H]^−^, 281.1169[M-H-4H_2_O]^−^, 190.0546[M-H-C_9_H_7_O_3_]^−^, 161.0285[M-H-C_7_H_12_O_6_]^−^, 134.0436[M-H-C_8_H_11_O_7_]^−^	Neochlorogenic acid	CML, IML	[39]
9	1.38	C_16_H_23_NO_7_	341.1458	341.1475	−4.7	342.1531[M + H]^+^, 261.1188[M + H-2H_2_O-C_2_H_5_O]^+^, 107.0492[M + H-Rha-C_3_H_6_NO_2_]^+^, 102.0550[M + H-Rha-C_6_H_4_]^+^	*O*-Ethyl-4-[(*α*-l-rhamnosyloxy)-benzyl]carbamate	CML, IML	[18]
10	1.47	C_7_H_6_O_4_	154.0271	154.0266	3.2	153.0215[M − H]^−^, 135.0211[M-H-H_2_O]^−^, 89.0340[M-H-H_2_O-HCOOH]^−^	3,4-Dihydroxy-benzoic acid	CML, IML	[40]
11	1.51	C_13_H_16_O_7_	300.0892	300.0896	−1.5	299.0819[M − H]^−^, 160.0351[M-H-H_2_O-C_7_H_5_O_2_]^−^, 90.0343[M-H-Glu-HCOOH]^−^	Benzaldehyde 4-*O-β*-glucoside	CML, IML	[41]
12 *	1.70	C_31_H_34_O_14_	630.1957	630.1949	1.3	675.1692[M + HCOO]^−^, 414.1127[M-H-Ph-CH_3_-C_7_H_7_O_2_]^−^, 353.0869[M-H-H_2_O-C_7_H_6_O-C_8_H_8_O_3_]^−^, 298.0797[M-H-CH_3_-C_17_H_16_O_6_]^−^, 222.0634[M-H-Ph-Glu-C_8_H_7_O_3_]^−^	Mudanpioside J	CML >> IML VIP: 2.73 *p* < 0.001	[42]
13	1.71	C_32_H_40_N_2_O_13_	660.2563	660.2530	4.6	705.2545[M + HCOO]^−^, 441.1367[M-H-Rha-C_2_H_3_-C_2_H_4_]^−^, 326.0797[M-H-Rha-C_11_H_9_N_2_]^−^, 263.0856[M-H-Rha-Glu-C_4_H_5_]^−^, 175.0444[M-H-Rha-Glu-C_10_H_6_N]^−^	N, *α*-l-Rhamnopyranosyl vincosamide	CML, IML	[43]
14 *	1.84	C_15_H_12_O_7_	304.0573	304.0583	−3.2	349.00618[M + HCOO]^−^, 285.0418[M-H-H_2_O]^−^, 162.0364[M-H-C_6_H_5_O_4_]^−^, 152.9691[M-H-CH_3_-C_8_H_8_O_2_]^−^, 132.0231[M-H-H_2_O-OCH_3_-C_7_H_6_O_2_]^−^, 130.0235[M-H-C_11_H_9_O_2_]^−^	Dihydroquercetin	CML VIP: 8.20 *p* < 0.001	[44]
15	2.07	C_16_H_18_O_9_	354.0951	354.0951	0.1	353.0878[M − H]^−^, 253.1035[M-H-3H_2_O-HCOOH]^−^, 190.0182[M-H-3H_2_O-C_6_H_5_O_2_]^−^, 144.0302[M-H-H_2_O-C_7_H_11_O_6_]^−^, 125.0251[M-H-H_2_O-HCOOH-C_9_H_8_O_3_]^−^	Chlorogenic acid	CML, IML	s
16	2.09	C_17_H_20_O_9_	368.1102	368.1107	−1.5	367.1029[M − H]^−^, 336.0902[M-H-OCH_3_]^−^, 295.1124[M-H-4H_2_O]^−^, 243.0591[M-H-CH_3_-C_6_H_5_O_2_]^−^, 189.0549[M-H-CH_3_-C_9_H_7_O_3_]^−^, 178.0346[M-H-C_8_H_13_O_5_]^−^	Methyl-3-caffeoylquinate	CML, IML	[45]
17	2.34	C_16_H_18_N_2_O_4_	302.1254	302.1267	−4.1	303.1327[M + H]^+^, 285.1232[M + H-H_2_O]^+^, 212.0983[M + H-H_2_O-C_2_H_5_O_2_]^+^, 176.0893[M + H-2H_2_O-C_3_H_7_O_3_]^+^	Tangutorid E	CML, IML	[45]
18	2.35	C_19_H_28_O_12_	448.1578	448.1581	−0.7	447.1505[M − H]^−^,417.0973[M-H-2CH_3_]^−^, 267.1031[M-H-Glu]^−^, 245.1016[M-H-OCOCH_3_-C_6_H_7_O_4_]^−^, 167.0480[M-H-Glu-C_4_H_4_O_3_]^−^	8-*O*-Acetylshanzhiside methyl ester	CML, IML	[46]
19	2.53	C_9_H_8_O_4_	180.0414	180.0423	−4.5	179.0335[M − H]^−^, 143.0430[M-H-2H_2_O]^−^, 133.0433[M-H-HCOOH]^−^, 108.0265[M-H-C_3_H_3_O_2_]^−^	Caffeic acid	CML, IML	s
20	2.62	C_16_H_18_N_2_O_4_	302.1257	302.1267	−3.1	303.1330[M + H]^+^, 285.1248[M + H-H_2_O]^+^, 194.0881[M + H-2H_2_O-C_2_H_5_O_2_]^+^, 194.0895[M + H-H_2_O-C_3_H_7_O_2_]^+^, 118.0799[M + H-H_2_O-C_11_H_7_N_2_]^+^	Tangutorid F	CML, IML	[45]
21	2.87	C_17_H_20_O_9_	368.1104	368.1107	−0.9	367.1031[M − H]^−^, 336.0931[M-H-OCH_3_]^−^, 203.0655[M-H-C_9_H_8_O_3_]^−^, 188.0545[M-H-CH_3_-C_9_H_8_O_3_]^−^, 151.0384[M-H-2H_2_O-C_9_H_8_O_4_]^−^	Methyl-4-caffeoylquinate	CML, IML	[45]
22	2.96	C_10_H_20_O_6_	236.1260	236.1261	0.3	259.1153[M + Na]^+^, 219.1322[M + H-H_2_O]^+^, 176.0465[M + H-H_2_O-C_3_H_7_]^+^, 164.0694[M + H-C_4_H_9_O]^+^	n-Butyl-*β-*d-fructopyranoside	CML, IML	[47]
23	3.00	C_18_H_26_O_10_	402.1536	402.1526	2.3	425.1428[M + Na]^+^, 296.1001[M + H-C_7_H_7_O]^+^, 253.1061[M + H-Xyl]^+^, 146.0584[M + H-Xyl-C_7_H_7_O]^+^, 73.0491[M + H-Glu-Xyl]^+^	Benzyl-*O-β*-d-xylopyranosyl-(1→6)-*β*-d-glucopyranoside	CML, IML	[18]
24	3.01	C_19_H_28_O_12_	448.1587	448.1581	1.5	447.1514[M − H]^−^,398.1453[M-H-H_2_O-OCH_3_]^−^, 378.1102[M-H-3H_2_O-CH_3_]^−^, 291.0974[M-H-C_8_H_12_O_3_]^−^, 267.1025[M-H-Glu]^−^, 193.0447[M-H-Glu-OCH_3_-COCH_3_]^−^	6-*O*-acetylshanzhiside methyl ester	CML, IML	[46]
25	3.04	C_14_H_17_NO_6_	295.1051	295.1056	−1.7	294.0978[M-H]^−^, 268.1025[M-H-CN]^−^, 162.0436[M-H-C_8_H_6_NO]^−^, 130.0390[M-H-Rha]^−^, 104.0286[M-H-Rha-CN]^−^	Niaziridin	CML, IML	[48]
26	3.13	C_9_H_8_O_3_	164.0474	164.0473	0.3	165.0544[M + H]^+^, 147.0444[M + H-H_2_O]^+^, 119.0483[M + H-HCOOH]^+^	*o*-Coumaric acid	CML, IML	[49]
27	3.23	C_27_H_30_O_15_	594.1590	594.1585	0.8	593.1517[M − H]^−^, 575.1371[M-H-H_2_O]^−^, 529.0871[M-H-H_2_O-Glu]^−^, 394.1305[M-H-2H_2_O-C_9_H_6_O_3_]^−^	Vicenin-2	CML, IML	[50]
28	3.32	C_9_H_8_O_3_	164.0471	164.0473	−1.4	165.0544[M + H]^+^, 147.0442[M + H-H_2_O]^+^, 119.0482[M + H-HCOOH]^+^, 107.0495[M + H-C_2_H_2_O_2_]^+^	*ρ*-Coumaric acid	CML, IML	[49]
29 *	3.34	C_31_H_34_O_14_	630.1943	630.1949	−0.8	675.1939[M + HCOO]^−^, 464.0735[M-H-2CH_3_-C_8_H_7_O_2_]^−^, 339.0923[M-H-H_2_O-C_7_H_5_O_2_-C_8_H_7_O_3_]^−^, 223.0599[M-H-C_7_H_7_O_2_-Glu benzoate]^−^, 163.0386[M-H-C_9_H_9_O_4_-Glu benzoate]^−^	6′-*O*-Benzoyl-4″-hydroxy-3″-methoxypaeoniflorin	CML >> IML VIP: 2.12 *p* < 0.001	[51]
30 *	3.35	C_16_H_18_O_8_	338.997	338.1002	−1.5	337.0930[M − H]^−^, 265.0787[M-H-4H_2_O]^−^, 173.0442[M-H-C_9_H_7_O_3_]^−^, 162.0386[M-H-C_7_H_11_O_5_]^−^, 127.0704[M-H-HCOOH-C_9_H_7_O_3_]^−^	3-*p*-Coumaroylquinic acid	CML >> IML VIP: 9.19 *p* < 0.001	[52]
31	3.47	C_27_H_30_O_15_	594.1589	594.1585	0.7	593.1516[M − H]^−^, 575.1396[M-H-H_2_O]^−^, 411.0869[M-H-H_2_O-Rha]^−^, 287.0536[M-H-H_2_O-Rha-C_6_H_4_O_3_]^−^, 125.0302[M-H-Rha-Glu-C_6_H_4_O_3_]^−^	Kaempferol-3-*O*-rutinoside	CML, IML	s
32	3.54	C_13_H_16_O_6_	268.0942	268.0947	−1.6	313.0924[M + HCOO]^−^, 213.0760[M-H-3H_2_O]^−^, 184.0768[M-H-3H_2_O-CHO]^−^, 147.0540[M-H-CH_3_-C_7_H_5_O]^−^, 103.0284[M-H-Rha]^−^	Benzaldehyde-4-*O-α*-l-rhamnopyranoside	CML, IML	[45]
33 *	3.55	C_16_H_14_O_7_	318.0746	318.0740	1.9	363.0747[M + HCOO]^−^, 208.0473[M-H-C_6_H_5_O_2_]^−^, 193.0273[M-H-CH_3_-C_6_H_5_O_2_]^−^, 133.0452[M-H-H_2_O-C_8_H_6_O_4_]^−^, 121.0284[M-H-C_9_H_8_O_5_]^−^	Padmatin	CML VIP: 3.75 *p* < 0.001	s
34	3.89	C_17_H_20_O_9_	368.1097	368.1107	−2.7	367.1025[M − H]^−^, 298.0387[M-H-3H_2_O-CH_3_]^−^, 288.1015[M-H-H_2_O-CH_3_-HCOOH]^−^, 192.0488[M-H-C_7_H_11_O_5_]^−^, 191.0629[M-H-C_10_H_8_O_3_]^−^	4-Feruloylquinic acid	CML, IML	a
35	4.02	C_14_H_17_NO_5_	279.1100	279.1107	−2.1	324.1082[M + HCOO]^−^, 188.0725[M-H-C_3_H_6_O]^−^, 147.0545[M-H-CH_3_-C_8_H_6_N]^−^, 114.0433[M-H-Rha]^−^, 88.0545[M-H-Rha-CN]^−^	Niazirin	CML, IML	[45]
36	4.05	C_27_H_30_O_17_	626.1487	626.1483	0.7	625.1414[M − H]^−^, 445.0853[M-H-Glu]^−^, 318.0205[M-H-Glu-H_2_O-C_6_H_5_O_2_]^−^, 324.1075[M-H-C_15_H_9_O_7_]^−^, 265.0333[M-H-2Glu]^−^, 275.0708[M-H-Glu-H_2_O-C_7_H_4_O_4_]^−^	QuerQuercetin-3-gentiobioside	CML, IML	a
37	4.14	C_15_H_14_O_6_	290.0784	290.0790	−1.8	335.0766[M + HCOO]^−^, 162.0243[M-H-H_2_O-C_6_H_5_O_2_]^−^, 138.0291[M-H-H_2_O-C_7_H_6_O_3_]^−^, 120.0283[M-H-C_8_H_9_O_4_]^−^, 79.0342[M-H-H_2_O-C_10_H_8_O_4_]^−^	Epicatechin	CML, IML	s
38 *	4.16	C_18_H_22_O_8_	366.1324	366.1315	2.5	411.1641[M + HCOO]^−^, 335.0765[M-H-2CH_3_]^−^, 232.0622[M-C_9_H_9_O]^−^, 173.0459[M-H-CH_3_-C_10_H_9_O_3_]^−^, 161.0243[M-CH_3_-Rha ethyl ester]^−^	3-*O*-acetyl-2-*O*-*p*-methoxycinnamoyl-*α*-l-rhamnopyranose	CML VIP: 2.69 *p* < 0.001	[53]
39	4.22	C_21_H_20_O_11_	448.0999	448.1006	−1.4	447.0926[M − H]^−^, 429.0850[M-H-H_2_O]^−^, 267.0395[M-H-Glu]^−^, 143.0288[M-H-Glu-C_6_H_4_O_3_]^−^	Astragalin	CML, IML	[54]
40	4.47	C_15_H_14_O_6_	290.0793	297.0790	0.6	335.0775[M + HCOO]^−^, 147.0436[M-H_2_O-C_6_H_4_O_3_]^−^, 137.0224[M-H-C_8_H_8_O_3_]^−^, 133.0295[M-H-H_2_O-C_7_H_6_O_3_]^−^, 90.0342[M-H-H_2_O-C_9_H_9_O_4_]^−^	Catechin	CML, IML	s
41	4.49	C_21_H_20_O_12_	464.0955	464.0949	−1.2	465.1022[M + H]^+^, 285.0485[M + H-Glu]^+^, 231.0678[M + H-Glu-3H_2_O]^+^, 149.0150[M + H-Glu-C_7_H_4_O_3_]^+^, 152.0154[M + H-Glu-C_8_H_5_O_2_]^+^	Hyperoside	CML, IML	s
42	4.50	C_27_H_30_O_17_	626.1475	626.1483	−1.3	625.1402[M − H]^−^, 516.1277[M-H-C_6_H_5_O_2_]^−^, 396.0689[M-H-Glu-H_2_O-CH_2_OH]^−^, 265.0264[M-H-2Glu]^−^, 132.9991[M-H-2Glu-C_8_H_5_O_2_]^−^	Quercetin-3-sophoroside	CML, IML	a
43	4.52	C_26_H_28_O_16_	596.1400	596.1377	3.8	595.1327[M − H]^−^, 265.0264[M-H-Glu-Xyl]^−^, 138.0156[M-H-Glu-Xyl-H_2_O-C_6_H_5_O_2_]^−^, 115.9991[M-H-Glu-Xyl-C_8_H_6_O_3_]^−^, 144.0485[M-H-Xyl-C_15_H_9_O_7_]^−^	Quercetin-3-*O-β*-d-xylopyranosyl-(1→2)-*β*-d-glucopyranoside	CML, IML	a
44	4.67	C_21_H_20_O_12_	464.0939	464.0955	−3.4	463.0866[M − H]^−^, 318.0758[M-H-2H_2_O-C_6_H_5_O_2_]^−^, 178.0513[M-H-C_15_H_9_O_6_]^−^, 159.0379[M-H-Glu-C_6_H_4_O_3_]^−^	Isoquercetin	CML, IML	[54]
45	4.68	C_27_H_30_O_14_	578.1635	578.1636	−0.2	579.1707[M + H]^+^, 543.1466[M + H-2H_2_O]^+^, 415.1130[M + H-Rha]^+^, 322.0748[M + H-Rha-C_6_H_5_O]^+^, 235.0580[M + H-Glu-Rha]^+^	Apigenin-7-*O*-rutinoside	CML, IML	[39]
46	4.71	C_27_H_30_O_15_	594.1596	594.1585	2.0	593.1524[M − H]^−^, 413.0899[M-H-Glu]^−^, 338.0756[M-H-Glu-H_2_O-C_2_HO_2_]^−^, 247.0305[M-H-Rha-Glu]^−^, 160.0677[M-H-Rha-C_15_H_9_O_5_]^−^	Kaempferol-3-*O-α*-l-rhamnoside-(1→4)-*β-*d-glucoside	CML, IML	a
47	4.82	C_26_H_34_O_11_	522.2118	522.2101	3.0	567.2100[M + HCOO]^−^, 461.2005[M-H-C_2_H_4_O]^−^, 341.1509[M-H-Glu]^−^, 401.1193[M-H-C_9_H_12_]^−^, 200.0871[M-H-Glu-H_2_O-C_7_H_7_O_2_]^−^, 134.0427[M-H-Glu-C_12_H_15_O_3_]^−^	Ligan glycoside A	CML, IML	b
48	4.85	C_26_H_28_O_14_	564.1475	564.1479	−0.7	565.1548[M + H]^+^, 418.1217[M + H-C_9_H_6_O_2_]^+^, 298.0909[M + H-Api-C_8_H_6_O]^+^, 180.0776[M + H-Api-C_15_H_9_O_4_]^+^, 147.0593[M + H-Glu-Api-C_6_H_3_O_2_]^+^	Apiin	CML, IML	[55]
49	4.95	C_15_H_10_O_7_	302.0429	302.0427	0.7	303.0501[M + H]^+^, 153.0162[M + H-C_8_H_6_O_3_]^+^, 151.0210[M + H-C_7_H_4_O_4_]^+^, 122.0388[M + H-C_9_H_6_O_4_]^+^	Quercetin	CML, IML	s
50 ^#^	4.97	C_27_H_30_O_16_	610.1537	610.1534	0.6	611.1652[M + H]^+^, 447.1016[M + H-Rha]^+^, 267.0509[M + H-Glu-Rha]^+^, 158.0289[M + H-Glu-Rha-C_6_H_5_O_2_]^+^, 131.0222[M + H-Glu-Rha-C_7_H_4_O_3_]^+^	Rutin	CML << IML VIP: 8.51 *p* < 0.001	s
51 ^#^	5.01	C_27_H_30_O_16_	610.1532	610.1534	−0.3	609.1472[M − H]^−^, 427.0974[M-H-Rha-H_2_O]^−^, 336.0683[M-H-Rha-C_6_H_5_O_2_]^−^, 265.0326[M-H-Glu-Rha]^−^, 132.0015[M-H-Glu-Rha-C_8_H_5_O_2_]^−^	Quercetin-3-rutinoside	CML << IML VIP: 13.28 *p* < 0.001	[56]
52	5.02	C_21_H_20_O_10_	432.1048	432.1056	−2.0	431.0975[M − H]^−^, 395.0746[M-H-2H_2_O]^−^, 338.0683[M-H-C_6_H_5_O]^−^, 251.0447[M-H-Glu]^−^, 100.0326[M-H-Glu-C_7_H_3_O_4_]^−^	Apigenin-8-C-glucoside	CML, IML	[39]
53	5.04	C_21_H_20_O_10_	432.1069	432.1056	2.7	477.1051[M + HCOO]^−^, 267.0464[M-H-Rha]^−^, 163.0701[M-H-C_15_H_8_O_5_]^−^, 115.0438[M-H-Rha-C_7_H_4_O_4_]^−^	Kaempherol-3-*O-α*-rhamnoside	CML, IML	[41]
54	5.08	C_15_H_19_NO_5_	293.1265	293.1263	0.5	294.1337[M + H]^+^, 131.0526[M + H-OCH_3_-C_8_H_6_NO]^+^, 99.0646[M + H-Rha-OCH_3_]^+^	4-(4 ′-*O*-methyl-*α-*l-rhamnosyloxy)benzyl nitrile	CML, IML	[18]
55 *	5.22	C_21_H_20_O_12_	464.0938	464.0955	−3.5	463.0880[M − H]^−^, 283.0502[M-H-Glu]^−^, 174.0278[M-H-Glu-C_6_H_5_O_2_]^−^, 150.0174[M-H-Glu-C_8_H_5_O_2_]^−^	Quercetin 3-*O-β*-d-glucopyranoside	CML >> IML VIP: 7.30 *p* < 0.001	[56]
56	5.48	C_15_H_14_O_5_	274.0837	274.0841	−1.2	319.0819[M + HCOO]^−^, 144.0281[M-H-2H_2_O-C_6_H_5_O]^−^, 137.0222[M-H-C_8_H_8_O_2_]^−^, 117.0329[M-H-H_2_O-C_7_H_6_O_3_]^−^, 92.0344[M-H-C_9_H_9_O_4_]^−^	(−)-Epiafzelechin	CML, IML	s
57	5.55	C_27_H_30_O_17_	626.1490	626.1483	1.1	625.1417[M − H]^−^, 571.1354[M-H-3H_2_O]^−^, 391.0807[M-H-Glu-3H_2_O]^−^, 303.0966[M-H-Glu-H_2_O-C_6_H_3_O_2_]^−^, 265.0399[M-H-2Glu]^−^	Quercetin-3,7-*O-β*-d-diglucopyranoside	CML, IML	a
58	5.65	C_24_H_22_O_15_	550.0957	550.0959	−0.2	549.0885[M − H]^−^, 445.0780[M-H-malonyl]^−^, 300.0267[M-H-malonyl-Glu]^−^, 160.0133[M-H-malonyl-Glu-C_6_H_4_O_3_]^−^	Quercetin-3-*O*-(6″-malonyl) glucoside	CML, IML	[49]
59 ^#^	5.67	C_27_H_30_O_15_	594.1579	594.1585	−1.0	595.1678[M + H]^+^, 448.1063[M + H-3H_2_O-C_6_H_5_O]^+^, 385.1335[M + H-Glu-C_2_HO]^+^, 304.0494[M + H-H_2_O-Glu-C_6_H_5_O]^+^, 142.0169[M + H-2Glu-C_6_H_5_O]^+^	Isovitexin-3″-*O*-glucopyranoside	CML << IML VIP: 5.63 *p* < 0.001	[57]
60 ^#^	5.69	C_30_H_26_O_13_	594.1355	594.1373	−3.1	593.1507[M − H]^−^, 484.1116[M-H-C_6_H_5_O_2_]^−^, 439.0848[M-H-C_7_H_6_O_4_]^−^, 286.0394[M-H-H_2_O-C_15_H_13_O_6_]^−^, 153.9989[M-H-C_23_H_20_O_9_]^−^	Procyanidins	CML << IML VIP: 7.04 *p* < 0.001	[58]
61	5.72	C_27_H_28_O_16_	608.1378	608.1377	0.0	607.1305[M − H]^−^, 543.1275[M-H-H_2_O-HCOOH]^−^, 504.0985[M-H-C_4_H_7_O_3_]^−^, 440.0889[M-H-H_2_O-C_5_H_9_O_3_]^−^, 462.0868[M-H-C_6_H_9_O_4_]^−^, 282.0267[M-H-Glu-C_6_H_9_O_4_]^−^	Quercetin-3-*O*-hydroxy methylglutaroyl galactoside	CML, IML	[59]
62 ^#^	5.84	C_28_H_32_O_16_	624.1703	624.1690	2.1	623.1611[M − H]^−^, 590.1383[M-H-H_2_O-CH_3_]^−^, 466.1438[M-H-2H_2_O-C_7_H_7_O_2_]^−^, 337.0986[M-H-Rha-C_6_H_4_O_3_]^−^, 281.0460[M-H-Glu-Rha]^−^	Isorhamnetin-3-*O*-rutinoside	CML << IML VIP: 2.90 *p* < 0.001	s
63*	5.92	C_21_H_20_O_11_	448.1003	448.1006	−0.7	447.0929[M − H]^−^, 267.0463[M-H-Glu]^−^, 227.0343[M-H-Glu-C_2_O]^−^, 134.0018[M-H-Glu-C_8_H_5_O_2_]^−^	Kaempferol-3-*O*-glucoside	CML >> IML VIP: 10.89 *p* < 0.001	[49]
64	6.07	C_16_H_12_O_7_	316.0581	316.0583	−0.6	317.0654[M + H]^+^, 302.0412[M + H-CH_3_]^+^, 299.0533[M + H-H_2_O]^+^, 152.0169[M + H-C_9_H_8_O_3_]^+^, 125.0388[M + H-C_10_H_8_O_4_]^+^	Isorhamnetin	CML, IML	s
65	6.16	C_23_H_22_O_13_	506.1068	506.1060	1.4	505.0995[M − H]^−^, 490.0815[M-H-CH_3_]^−^, 428.0988[M-H-H_2_O-OCOCH_3_]^−^, 317.0980[M-H-2H_2_O-C_7_H_4_O_4_]^−^, 283.0198[M-H-Glu ethyl ester]^−^	Quercetin-3-*O*-(6″-*O*-acetyl)-*β*-d-glucopyranoside	CML, IML	[49]
66	6.24	C_9_H_16_O_4_	188.1045	188.1049	2.1	187.0965[M − H]^−^, 141.1105[M-H-HCOOH]^−^, 123.0957[M-H-H_2_O-HCOOH]^−^, 112.0644[M-H-H_2_O-C_3_H_5_O]^−^	Azelaic acid	CML, IML	s
67	6.28	C_22_H_22_O_9_	430.1242	430.1264	−4.7	475.1224[M + HCOO]^−^, 288.0536[M-H-H_2_O-C_7_H_7_O_2_]^−^, 244.0915[M-H-2H_2_O-C_9_H_8_O_2_]^−^, 143.0398[M-H-CH_3_-Rha-C_6_H_3_O_2_]^−^, 130.0289[M-H-Rha-C_7_H_3_O_3_]^−^	Chryseriol-7-*O*-rhamnoside	CML, IML	[39]
68	6.43	C_36_H_36_O_18_	756.1900	756.1902	−0.3	755.1827[M − H]^−^, 737.1844[M-H-H_2_O]^−^, 575.1386[M-H-Glu]^−^, 427.0933[M-H-Glu-C_9_H_7_O_2_]^−^, 405.0904[M-H-Glu-H_2_O-C_7_H_4_O_4_]^−^, 247.0320[M-H-2Glu-C_9_H_7_O_2_]^−^	Allivictoside A	CML, IML	b
69 *	6.44	C_34_H_24_O_10_	592.1387	592.1370	3.0	593.1639[M + H]^+^, 483.1521[M + H-H_2_O-C_6_H_4_O]^+^, 266.0696[M + H-C_8_H_5_O_2_-C_10_H_10_O_2_]^+^, 241.0502[M + H-C_20_H_16_O_6_]^+^, 134.0267[M + H-C_26_H_19_O_8_]^+^	Mulberrofuran Q	CML >> IML VIP: 5.76 *p* < 0.001	[59]
70	6.46	C_24_H_22_O_14_	534.1014	534.1010	0.8	533.0941[M − H]^−^, 447.0920[M-H-malonyl]^−^, 323.0962[M-H-malonyl-C_6_H_4_O_3_]^−^, 284.0320[M-H-malonyl-Glu]^−^	Kaempferol-3-*O*-(6″-malonyl) glucoside	CML, IML	[49]
71	6.74	C_22_H_20_O_12_	476.0964	476.0955	1.8	475.0891[M − H]^−^, 444.0726[M-H-OCH_3_]^−^, 351.0892[M-H-C_6_H_4_O_3_]^−^, 283.0394[M-H-methyl glucuronate]^−^, 172.0452[M-H-H_2_O-C_15_H_9_O_6_]^−^	Kaempferol-3-*O-β*-d-glucuronide-6″-methyl ester	CML, IML	[60]
72	6.98	C_21_H_20_O_10_	432.1052	432.1056	−1.1	431.0979[M − H]^−^, 267.0327[M-H-Rha]^−^, 249.0447[M-H-Rha-H_2_O]^−^, 157.9997[M-H-Rha-C_6_H_6_O_2_]^−^	Kaempferol-7-*O-α*-l-rhamnoside	CML, IML	[61]
73	6.99	C_15_H_10_O_6_	286.0484	286.0477	2.4	287.0557[M + H]^+^, 153.0167[M + H-C_8_H_6_O_2_]^+^, 135.0583[M + H-C_7_H_4_O_4_]^+^, 124.0385[M + H-C_9_H_6_O_3_]^+^	Orobol	CML, IML	[62]
74 *	7.01	C_23_H_22_O_12_	490.1108	490.1111	−0.6	489.1039[M − H]^−^, 446.1001[M-H-COCH_3_]^−^, 267.0323[M-H-Glu ethyl ester]^−^, 143.0443[M-H-Glu ethyl ester-C_6_H_4_O_3_]^−^	3-*O*-(6″-*O*-acetyl)-*β-*d-glucopyranside	CML >> IML VIP: 6.15 *p* < 0.001	[63]
75	7.80	C_16_H_19_NO_6_	321.1213	321.1212	0.2	366.1195[M + HCOO]^−^, 249.0617[M-H-OCH _3_-C_2_H_2_N]^−^, 189.0517[M-H-CH_3_-C_8_H_6_N]^−^, 97.0370[M-H-Rha-C_2_H_3_O_2_]^−^	Niazirinin	CML, IML	[45]
76	7.86	C_15_H_10_O_6_	286.0477	286.0473	−1.7	285.0400[M − H]^−^, 121.0377[M-H-C_6_H_4_O_3_]^−^, 183.0010[M-H-C_8_H_6_]^−^, 133.0415[M-H-C_7_H_4_O_4_]^−^, 108.0280[M-H-C_9_H_5_O_4_]^−^	Luteolin	CML, IML	s
77	7.89	C_15_H_10_O_7_	302.0408	302.0427	−1.8	301.0336[M − H]^−^, 244.0329[M-H-C_2_HO_2_]^−^, 190.0130[M-H-H_2_O-C_6_H_5_O]^−^, 133.0269[M-H-C_7_H_4_O_5_]^−^, 92.0343[M-H-C_9_H_5_O_6_]^−^	6-Hydroxykaempferol	CML, IML	[64]
78 ^#^	7.89	C_28_H_32_O_17_	640.1653	640.1639	2.1	663.3153[M + Na]^+^, 443.0906[M + H-Glu-H_2_O]^+^, 281.0480[M + H-2Glu]^+^, 266.0487[M + H-2Glu-CH_3_]^+^, 158.0315[M + H-2Glu-C_7_H_7_O_2_]^+^	Isorhamnetin 3-*O-β*-gentiobioside	CML << IML VIP: 4.79 *p* < 0.001	a
79 *	8.01	C_15_H_10_O_6_	286.0479	286.0477	0.6	287.0567[M + H]^+^, 163.0580[M + H-C_6_H_4_O_3_]^+^, 147.0435[M + H-C_6_H_4_O_4_]^+^, 124.0384[M + H-C_9_H_6_O_3_]^+^	Scutellarein	CML >> IML VIP: 3.69 *p* < 0.001	[65]
80 *	8.02	C_25_H_24_O_12_	516.1266	516.1268	−0.3	515.1203[M − H]^−^, 451.1465[M-H-H_2_O-HCOOH]^−^, 326.0487[M-H-3H_2_O-C_8_H_7_O_2_]^−^, 219.0638[M-H-2C_9_H_7_O_2_]^−^, 143.0279[M-H-H_2_O-C_16_H_18_O_9_]^−^	1,3-Dicaffeoylquinic acid	CML >> IML VIP: 4.25 *p* < 0.001	s
81 ^#^	8.26	C_20_H_26_O_9_	410.1573	410.1577	−1.0	409.1504[M − H]^−^, 336.0817[M-H-C_4_H_9_O]^−^, 251.1394[M-H-2H_2_O-C_7_H_6_O_2_]^−^, 202.0639[M-H-C_3_H_7_-C_9_H_7_O_3_]^−^, 134.0437[M-H-C_12_H_19_O_7_]^−^	5-*O*-Caffeoylquinic acid butyl ester	CML << IML VIP: 4.72 *p* < 0.001	a
82 ^#^	8.52	C_38_H_48_O_19_	808.2757	808.2790	−3.9	853.2706[M + HCOO]^−^, 700.2173[M-H-C_7_H_7_O]^−^, 572.1546[M-H-C_4_H_7_-Glu]^−^, 438.1262[M-H-C_4_H_7_-Rha-Ara]^−^, 274.1182[M-H-Glu-Ara-C_12_H_11_O_3_]^−^	7-(*α*-L-Galactopyranosyloxy)-5-hydroxy-2-(4-methoxyphenyl)-8-(3-methyl-2-buten-1-yl)-4-oxo-4*H*-chromen	IML VIP: 2.08 *p* < 0.001	b
83	9.08	C_30_H_34_O_15_	634.1872	634.1898	−3.7	679.1854[M + HCOO]^−^, 600.1579[M-H-H_2_O-CH_3_]^−^, 454.10677[M-H-Rha-CH_3_]^−^, 411.0997[M-H-CH_3_-Rha ethyl ester]^−^, 334.0931[M-H-Rha-C_7_H_3_O_3_]^−^, 296.0665[M-H-C_9_H_7_O-Rha ethyl ester]^−^	Kaempferol-3-*O-α*-l-(4-*O*-acetyl)-rhamnosyl-7-*O-α*-l-rhamnoside	CML, IML	[41]
84 *	9.23	C_15_H_10_O_6_	286.0473	286.0477	−1.7	285.0435[M − H]^−^, 228.0285[M-H-C_2_HO_2_]^−^, 161.0377[M-H-C_6_H_4_O_3_]^−^, 151.0010[M-H-C_8_H_6_O_2_]^−^	Kaempferol	CML >> IML VIP: 4.99 *p* < 0.001	s
85	9.31	C_14_H_17_NO_5_S	311.0821	311.0827	−1.9	356.0803[M + HCOO]^−^, 252.0915[M-H-NCS]^−^, 162.0681[M-H-C_8_H_6_NS]^−^, 88.0495[M-H-Rha-NCS]^−^	4-[(*α*-l-rhamnosyloxy) benzyl] Isothiocyanate	CML, IML	[66]
86	9.47	C_16_H_12_O_7_	316.0575	316.0583	−2.4	315.0503[M − H]^−^, 300.0268[M-H-CH_3_]^−^, 282.0400[M-H-H_2_O-CH_3_]^−^, 191.0163[M-H-CH_3_-C_6_H_5_O_2_]^−^, 165.0069[M-H-C_8_H_6_O_3_]^−^	Rhamnetin	CML, IML	[18]
87	9.54	C_12_H_16_O_4_	224.1038	224.1049	−4.8	223.0965[M − H]^−^, 205.1027[M-H-H_2_O]^−^, 135.0421[M-H-C_4_H_8_O_2_]^−^, 123.0964[M-H-C_4_H_4_O_3_]^−^, 87.0295[M-H-C_8_H_8_O_2_]^−^	3-Butylidene-4,5,6,7-tetrahydro-6,7-dihydroxy-1(3H)-isobenzofuranone	CML, IML	[67]
88	9.69	C_30_H_36_O_4_	460.2612	460.2614	−0.4	505.2629[M + HCOO]^−^, 444.2249[M-H-CH_3_]^−^, 372.1847[M-H-H_2_O-C_5_H_9_]^−^, 240.1718[M-H-CH_3_-C_12_H_12_O_3_]^−^, 139.0822[M-H-H_2_O-C_4_H_7_O-C_14_H_15_O_3_]^−^	Sophoranone	CML, IML	a
89	9.94	C_15_H_10_O_6_	286.0480	286.0477	1.1	287.0553[M + H]^+^, 153.0163[M + H-C_8_H_6_O_2_]^+^, 135.0449[M + H-C_7_H_4_O_4_]^+^, 124.0382[M + H-C_9_H_6_O_3_]^+^, 110.0281[M + H-C_9_H_5_O_4_]^+^	5,7,2′,5′-Tetrahydroxyflavone	CML, IML	b
90	9.98	C_27_H_28_O_12_	544.1583	544.1581	0.4	589.1565[M + HCOO]^−^, 375.1260[M-H-C_6_H_5_O_2_-C_2_H_3_O_2_]^−^, 328.0465[M-H-H_2_O-2OCH_3_-C_8_H_7_O_2_]^−^, 244.0508[M-H-C_9_H_7_O_3_-C_8_H_7_O_2_]^−^, 153.0016[M-H-2OCH_3_-2C_9_H_7_O_3_]^−^	1-*O*-methyl-3,5-*O*-dicaffeoylquinic acid methyl ester	CML, IML	a
91	10.10	C_29_H_32_O_15_	620.1741	620.1758	2.6	621.1830[M + H]^+^, 507.1514[M + H-C_9_H_6_]^+^, 310.0986[M + H-Rha-C_9_H_6_O_2_]^+^, 147.0658[M + H-Glu ethyl ester-C_15_H_9_O_4_]^+^	Apigenin-7-O-*α*-l-rhamnopyranosyl(1 → 4)-6″-O-acetyl-*β*-d-glucopyranoside	CML, IML	[68]
92	10.30	C_18_H_34_O_5_	330.2405	330.2406	−0.4	329.2332[M − H]^−^, 293.2084[M-H-2H_2_O]^−^, 226.1434[M-H-H_2_O-C_6_H_13_]^−^, 212.1325[M-H-HCOOH-C_5_H_11_]^−^, 168.1004[M-H-H_2_O-C_9_H_19_O]^−^, 137.1117[M-H-H_2_O-C_3_H_7_-C_6_H_11_O_3_]^−^	Sanleng acid	CML, IML	[26]
93 *	10.39	C_10_H_12_O_2_	164.0837	164.0831	−2.8	209.1118[M + HCOO]^−^, 122.0453[M-H-C_3_H_5_]^−^, 105.0495[M-H-OCH_3_-C_2_H_3_]^−^	Eugenol	CML >> IML VIP: 2.57 *p* < 0.001	s
94	10.52	C_15_H_10_O_6_	286.0477	286.0478	0.1	287.0550[M + H]^+^, 256.0427[M + H-OCH_3_]^+^, 167.0167[M + H-C_7_H_4_O_2_]^+^, 137.0227[M + H-C_8_H_6_O_3_]^+^, 121.0434[M + H-C_8_H_6_O_4_]^+^	1,7-Dihydroxy-2,3-methylenedioxyxanthone	CML, IML	b
95	10.67	C_15_H_30_O_2_	242.2241	242.2246	−1.6	287.2223[M + HCOO]^−^, 170.1211[M-H-C_5_H_11_]^−^, 153.1120[M-H-OCH_3_-C_4_H_9_]^−^, 97.0818[M-H-OCH_3_-C_8_H_17_]^−^, 69.0512[M-H-OCH_3_-C_10_H_21_]^−^	Methyl myristate	CML, IML	s
96	10.73	C_15_H_10_O_6_	286.0480	286.0477	0.8	287.0552[M + H]^+^, 153.0171[M + H-C_8_H_6_O_2_]^+^, 124.0390[M + H-C_9_H_6_O_3_]^+^, 110.0285[M + H-C_9_H_5_O_4_]^+^	2′-Hydroxygenistein	CML, IML	s
97 ^#^	11.22	C_18_H_34_O_5_	330.2393	330.2406	−4.0	329.2320[M − H]^−^, 213.1323[M-H-C_2_H_5_-C_4_H_7_O_2_]^−^, 208.1038[M-H-2H_2_O-C_6_H_13_]^−^, 183.1403[M-H-H_2_O-C_7_H_13_O_2_]^−^, 170.1223[M-H-C_8_H_15_O_3_]^−^	Tianshic acid	CML << IML VIP: 1.52 *p* < 0.001	[69]
98	11.35	C_16_H_19_NO_6_S	353.0930	353.0933	−0.8	398.0912[M + HCOO]^−^, 262.1963[M-H-H_2_O-C_2_H_2_NS]^−^, 236.0926[M-H-CH_3_-COCH_3_-C_2_H_2_S]^−^, 150.0741[M-H-C_2_H_2_NS-C_6_H_10_O_3_]^−^	4-[(4′-*O*-acetyl-*α*-l-rhamnosyloxy)benzyl]isothiocyanate	CML, IML	[66]
99	11.86	C_12_H_14_O_2_	190.0988	190.0994	−2.4	235.0970[M + HCOO]^−^, 146.0415[M-H-C3H7]^−^, 132.0273[M-H-C4H9]^−^, 113.0743[M-H-C6H4]^−^	3-n-Butylphthalide	CML, IML	a
100 ^#^	11.95	C_32_H_50_O_14_	658.3196	658.3201	−0.8	681.2695[M + Na]^+^, 617.2802[M + H-C_3_H_6_]^+^, 448.2238[M + H-Glu-CH_2_OH]^+^, 397.2184[M + H-Glu-C_6_H_10_]^+^, 203.0848[M + H-2Glu-C_7_H_12_]^+^	Ajugaside A	CML << IML VIP: 3.40 *p* < 0.001	[70]
101	12.01	C_30_H_40_O_12_	592.2544	592.2520	4.1	591.2471[M − H]^−^, 561.2176[M-H-2CH_3_]^−^, 365.1552[M-H-CH_3_-OCH_3_-Glu]^−^, 315.1166[M-H-C_16_H_20_O_4_]^−^, 211.1134[M-H-Glu-C_11_H_12_O_3_]^−^	Syringaresinolmono-*β*-d-glucoside	CML, IML	[71]
102	12.21	C_12_H_14_O_4_	222.0883	222.0892	−3.9	221.0811[M − H]^−^, 160.0546[M-H-OC_2_H_5_]^−^, 119.0282[M-H-C_3_H_5_O_2_-C_2_H_5_]^−^,	Diethyl phthalate	CML, IML	[67]
103	12.22	C_16_H_19_NO_6_S	353.0930	353.0933	−0.8	398.0912[M + HCOO]^−^, 265.0805[M-H-COCH_3_-CS]^−^, 161.0338[M-H-COCH_3_-C_8_H_6_NS]^−^, 101.0359[M-H-Rha-COCH_3_-CS]^−^	4-[(2′-*O*-acetyl-*α*-l-rhamnosyloxy) benzyl] Isothiocyanate	CML, IML	[49]
104 ^#^	12.98	C_20_H_23_N_7_O_7_	473.1668	473.1659	1.9	472.1665[M − H]^−^, 423.1223[M-H-H_2_O-NH_2_-NH]^−^, 383.1668[M-H-H_2_O-CHO-CH_2_N_2_]^−^, 351.0896[M-H-H_2_O-HCOOH-CH_3_N_3_]^−^, 164.0386[M-H-CHO-CH_3_N-C_12_H_12_NO_5_]^−^	Folinic acid	CML << IML VIP: 3.43 *p* < 0.001	a
105	13.10	C_21_H_38_O_4_	354.2757	354.2770	−3.3	399.2739[M + HCOO]^−^, 324.2187[M-H-C_2_H_5_]^−^, 238.1479[M-H-H_2_O-C_7_H_13_]^−^, 202.1101[M-H-C_11_H_19_]^−^, 151.1152[M-H-C_8_H_15_-C_3_H_7_O_3_]	2-Monolinolein	CML, IML	[26]
106	13.29	C_35_H_52_O_14_	696.3375	696.3357	2.4	741.3357[M + HCOO]^−^, 571.2800[M-H-C_7_H_8_O_2_]^−^, 433.2317[M-H-Glu-2H_2_O-HCOOH]^−^, 366.2232[M-H-Glu-Ribose]^−^, 303.2063[M-H-Glu-H_2_O-C_11_H_14_O_3_]^−^	Erysimosole	CML, IML	b
107	13.40	C_21_H_31_NO_10_S	489.1682	489.1669	2.8	488.1610[M − H]^−^, 473.1683[M-H-CH_3_]^−^, 308.1280[M-H-Glu]^−^, 293.0912[M-H-CH_3_-Glu]^−^, 218.0984[M-H-Glu-H_2_O-C_2_H_2_NS]^−^	4-[(*β*-d-glucopyranosyl-1-4-*α-*l-rhamnopyranosyloxy) benzyl] Isothiocyanate	CML, IML	a
108	13.52	C_13_H_16_O_3_	220.1095	220.1099	−1.9	219.1023[M − H]^−^, 164.0378[M-H-C_4_H_7_]^−^, 145.0326[M-H-OCH_3_-C_3_H_7_]^−^	4-(1-Oxopentyl)-methyl ester,Benzoic acid	CML, IML	a
109	14.68	C_18_H_34_O_4_	314.2443	314.2457	−4.6	313.2370[M − H]^−^, 199.1107[M-H-2C_4_H_9_]^−^, 184.1370[M-H-C_7_H_13_O_2_]^−^, 155.1048[M-H-C_9_H_17_O_2_]^−^, 125.1126[M-H-C_4_H_9_O-C_6_H_11_O_2_]^−^	Dibutyl sebacate	CML, IML	s
110	15.87	C_18_H_28_O_2_	276.2099	276.2089	3.3	277.2171[M + H]^+^, 150.1315[M + H-C_7_H_11_O_2_]^+^, 136.1167[M + H-C_8_H_13_O_2_]^+^, 107.0704[M + H-C_5_H_9_-C_5_H_9_O_2_]^+^, 95.0708[M + H-C_4_H_7_-C_7_H_11_O_2_]^+^	Parinaric acid	CML, IML	s
111	15.93	C_15_H_22_O_4_	266.1506	266.1518	−3.8	265.1488[M − H]^−^, 247.1494[M-H-H_2_O]^−^, 211.1328[M-H-C_3_H_2_O]^−^, 180.1365[M-H-CH_3_-C_3_H_2_O_2_]^−^, 169.1007[M-H-C_3_H_6_-C_3_H_2_O]^−^, 133.1009[M-H-H_2_O-C_5_H_6_O_3_]^−^	4*α*,6*α*-Dihydroxyeud-esman-8*β*,12-olide	CML, IML	a
112	16.00	C17H34O2	270.2556	270.2559	−0.9	315.2538[M + HCOO]^−^, 254.2163[M-H-CH_3_]^−^, 139.1285[M-H-OCH_3-_C_7_H_15_]^−^, 125.1118[M-H-OCH_3-_C_8_H_17_]^−^	Methyl palmitate	CML, IML	s
113	16.49	C_18_H_30_O_4_	310.2142	310.2144	−0.6	309.2069[M − H]^−^, 291.1964[M-H-H_2_O]^−^, 245.2069[M-H-H_2_O-HCOOH]^−^, 208.1397[M-H-C_5_H_9_O_2_]^−^, 198.1177[M-H-C_7_H_11_O]^−^, 135.0958[M-H-C_9_H_17_O_3_]^−^	9,16-Dihydroxy-10,12,14-octadecatrienoic acid	CML, IML	b
114	17.88	C_16_H_30_O_2_	254.2259	254.2246	4.8	277.2151[M + Na]^+^, 237.2359[M + H-H_2_O]^+^, 97.1016[M + H-C_2_H_5_-C_7_H_13_O_2_]^+^, 88.0605[M + H-C_12_H_23_]^+^, 69.0716[M + H-C_4_H_9_-C_7_H_13_O_2_]^+^	Palmitoleic acid	CML, IML	[32]
115 *	18.44	C_18_H_30_O_3_	294.2198	294.2195	0.9	293.2087[M − H]^−^, 275.2009[M-H-H_2_O]^−^, 247.2242[M-H-HCOOH]^−^, 232.1683[M-H-H_2_O-C_3_H_7_]^−^, 152.1063[M-H-H_2_O-C_9_H_15_]^−^	(*E,E*)-9-Oxooctadeca-10,12-dienoic acid	CML >> IML VIP: 2.45 *p* < 0.001	[30]
116 *	18.56	C_18_H_34_O_3_	298.2494	298.2508	−4.6	297.2440[M − H]^−^, 279.2478[M-H-H_2_O]^−^, 224.1515[M-H-H_2_O-C_4_H_7_]^−^, 139.1260[M-H-C_2_H_5_-C_7_H_13_O_2_]^−^, 139.1113[M-H-C_3_H_7_-C_6_H_11_O_2_]^−^	Ricinoleic acid	CML >> IML VIP: 3.12 *p* < 0.001	s
117	19.27	C_18_H_32_O_3_	296.2339	296.2351	−4.3	295.2266[M − H]^−^, 266.1996[M-H-C_2_H_5_]^−^, 249.2382[M-H-HCOOH]^−^, 184.1156[M-H-C_8_H_15_]^−^, 152.1412[M-H-HCOOH-C_7_H_13_]^−^, 124.0960[M-H-H_2_O-C_10_H_17_O]^−^	Coronaric acid	CML, IML	[26]
118	19.71	C_18_H_34_O_2_	282.2558	282.2559	−0.4	283.2631[M + H]^+^, 97.1020[M + H-C_5_H_11_-C_6_H_11_O_2_]^+^, 86.1024[M + H-C_12_H_21_O_2_]^+^, 72.0876[M + H-C_13_H_23_O_2_]^+^	Oleic acid	CML, IML	s
119 ^#^	19.95	C_21_H_36_O_4_	352.2620	352.2614	1.8	353.2701[M + H]^+^, 335.2693[M + H-H_2_O]^+^, 214.2202[M + H-C_3_H_7_O_3_]^+^, 150.1320[M + H-C_10_H_19_O_4_]^+^, 123.1012[M + H-C_5_H_9_-C_7_H_13_O_4_]^+^, 83.0715[M + H-C_7_H_11_-C_8_H_15_O_4_]^+^	1-Linolenoylglycerol	CML << IML VIP: 5.63 *p* < 0.001	a
120	21.54	C_18_H_30_O_2_	278.2237	278.2246	−3.0	277.2165[M − H]^−^, 182.1234[M-H-C_7_H_11_]^−^, 168.1230[M-H-C_8_H_13_]^−^, 110.0795[M-H-C_11_H_17_-H_2_O]^−^	Linolenic acid	CML, IML	s
121 *	21.70	C_15_H_30_O	226.2300	226.2297	1.3	271.2274[M + HCOO]^−^, 164.1118[M-H-C_6_H_13_]^−^, 108.0499[M-H-C_10_H_21_]^−^	n-Pentadecanal	CML >> IML VIP: 4.02 *p* < 0.001	[32]
122 *	24.46	C_17_H_34_O	254.2616	254.2610	2.1	299.2594[M + HCOO]^−^, 248.2224[M-H-C_2_H_5_]^−^, 122.0654[M-H-C_11_H_23_]^−^, 94.0506[M-H-C_13_H_27_]^−^	n-Heptadecanal	CML >> IML VIP: 3.12 *p* < 0.001	[32]

* Characteristic component in CML; ^#^ characteristic component in IML; ^s^ identified with standard; ^a^ compared with spectral data obtained from Wiley Subscription Services, Inc. (USA); ^b^ compared with NIST Chemistry WebBook.

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
