# Peer review of "Comparative Analysis of Chemical Constituents of *Moringa oleifera* Leaves from China and India by Ultra-Performance Liquid Chromatography Coupled with Quadrupole-Time-Of-Flight Mass Spectrometry"

_molecules, 2019, doi:10.3390/molecules24050942_

Round 1
Reviewer 1 Report
This is an interesting comparative study of Moringa oleifera.
I really like the short "Introduction" and the detailed description of the methods.
However, I recommend you improve the "Discussion": the results must be discussed and compared to others and you must provide ideas and thoughts, including limitations and future perspectives to the end.
Author Response
Response to Reviewer 1 Comments and Suggestions
Point 1: I recommend you improve the "Discussion": the results must be discussed and compared to others and you must provide ideas and thoughts, including limitations and future perspectives to the end.
Response 1: The "Discussion" has been improved including discussing results, comparing to others, providing some ideas and thoughts such as limitations and future perspectives to the end.
Reviewer 2 Report
In my opinion the manuscript molecules-457300 entitled “Comparative UPLC-QTOF-MSE analysis of chemical constituents of Moringa oleifera leaves from China and India” is suitable for publication in Molecules after major revision. Although the manuscript has some positive components (the number of identified compounds, potential interest) it has particular drawbacks which have to be corrected, namely:
1) Title should not contain abbreviations.
2) The authors should consult the Microsoft Word template to prepare the manuscript, since the current manuscript did not respect the molecules template. Moreover, Fig. and Tab. should be Figure and Table, and Figure 4A, 4B, and 4C should be Figure 4(a), 4(b), and 4(c).
3) All abbreviations should be described when used for the first time. After this, should be adopted through the manuscript. In addition, the abbreviations must be standardized, in this sense, the abbreviation of retention time should be RT and not tR as in Table 2.
4) The authors should pay attention to units, sometimes use µl other µL.
5) The Discussion is poor and not very meaningful. Emphasize the most relevant findings and try to give it more meaning, emphasize the novelty of your data.
6) Organic acid easters? This is correct? The reason for this question is because it appears several times in the manuscript.
7) Figure 3. The symbol not correspond to the legend. Please identify the ESI+ and ESI-, and the respective loading plots.
8) The conclusions are very similar to the abstract.
Author Response
Response to Reviewer 2 Comments and Suggestions
Point 1: Title should not contain abbreviations.
Response 1: The title has been changed to“Comparative analysis of chemical constituents of Moringa oleifera leaves from China and India by ultra-performance liquid chromatography coupled with quadrupole-time-of-flight mass spectrometry”
Point 2: The authors should consult the Microsoft Word template to prepare the manuscript, since the current manuscript did not respect the molecules template. Moreover, Fig. and Tab. should be Figure and Table, and Figure 4A, 4B, and 4C should be Figure 4(a), 4(b), and 4(c).
Response 2: The manuscript has been modified according to the above suggestion.
Point 3: All abbreviations should be described when used for the first time. After this, should be adopted through the manuscript. In addition, the abbreviations must be standardized, in this sense, the abbreviation of retention time should be RT and not tR as in Table 2.
Response 3: We have checked all the abbreviations in the article, and already added description when they were used for the first time. In addition, we have changed tR into RT to meet the standard of abbreviations.
Point 4: The authors should pay attention to units, sometimes use µl other µL.
Response 4: We have unified the units into µL.
Point 5: The Discussion is poor and not very meaningful. Emphasize the most relevant findings and try to give it more meaning, emphasize the novelty of your data.
Response 5: It is true that the discussion is poor and not very meaningful. Some relevant finding and the novelty of our data has been added in this section.
Point 6: Organic acid easters? This is correct? The reason for this question is because it appears several times in the manuscript.
Response 6: It is really the typo error in the manuscript. The text "organic acid esters" appeared in Figure 2, line 17, line 165 and line 273 were the right one. So, "organic acid easters" has been modified to "organic acid esters" in the revised manuscirpt.
Point 7: Figure 3. The symbol not correspond to the legend. Please identify the ESI+ and ESI-, and the respective loading plots.
Response 7: Sorry about this. The "ESI+" and "ESI-" were hided in the text boxes. And in the revised manuscript, it has been displayed completely.
Point 8: The conclusions are very similar to the abstract.
Response 8: The conclusions has been revised in order to avoid similarity to abstract.
Round 2
Reviewer 2 Report
All correction suggested by the reviewer was performed in the manuscript. For this reason, the manuscript should be accepted in the present form.